# A Prospective Study on the Roles of the Lymphocyte-to-Monocyte Ratio (LMR), Neutrophil-to-Lymphocyte Ratio (NLR), and Platelet-to-Lymphocyte Ratio (PLR) in Patients with Locally Advanced Rectal Cancer

**DOI:** 10.3390/biomedicines11113048

**Published:** 2023-11-14

**Authors:** Cieszymierz Gawiński, Andrzej Mróz, Katarzyna Roszkowska-Purska, Iwona Sosnowska, Edyta Derezińska-Wołek, Wojciech Michalski, Lucjan Wyrwicz

**Affiliations:** 1Department of Oncology and Radiotherapy, M. Skłodowska-Curie National Research Institute of Oncology, ul. Wawelska 15, 02-034 Warsaw, Poland; lucjan.wyrwicz@coi.pl; 2Department of Pathology, M. Skłodowska-Curie National Research Institute of Oncology, ul. Roentgena 5, 02-781 Warsaw, Poland; andrzej.mroz@coi.pl (A.M.); iwona.sosnowska@coi.pl (I.S.); edyta.derezinska-wolek@coi.pl (E.D.-W.); 3Department of Pathology, M. Skłodowska-Curie National Research Institute of Oncology, ul. Wawelska 15, 02-034 Warsaw, Poland; katarzyna.roszkowska-purska@coi.pl; 4Department of Computation Oncology, M. Skłodowska-Curie National Research Institute of Oncology, ul. Roentgena 5, 02-781 Warsaw, Poland; wojciech.michalski@coi.pl

**Keywords:** LMR (lymphocyte-to-monocyte ratio), NLR (neutrophil-to-lymphocyte ratio), PLR (platelet-to-lymphocyte ratio), inflammatory infiltrate, CPS (combined positive score), rectal cancer

## Abstract

Rectal cancer constitutes over one-third of all colorectal cancers (CRCs) and is one of the leading causes of cancer-related deaths in developed countries. In order to identify high-risk patients and better adjust therapies, new markers are needed. Systemic inflammatory response (SIR) markers such as LMR, NLR, and PLR have proven to be highly prognostic in many malignancies, including CRC; however, their roles in locally advanced rectal cancer (LARC) are conflicting and lack proper validation. Sixty well-selected patients with LARC treated at the Maria Sklodowska-Curie National Research Institute of Oncology in Warsaw, Poland, between August 2017 and December 2020 were prospectively enrolled in this study. The reproducibility of the pre-treatment levels of the SIR markers, their correlations with clinicopathological characteristics, and their prognostic value were evaluated. There was a significant positive correlation between LMR and cancer-related inflammatory infiltrate (r = 0.38, *p* = 0.044) and PD-L1 expression in tumor cells, lymphocytes, and macrophages (combined positive score (CPS)) (r = 0.45, *p* = 0.016). The PLR level was correlated with nodal involvement (*p* = 0.033). The SIR markers proved to be only moderately reproducible and had no significant prognostic value. In conclusion, the LMR was associated with local cancer-related inflammation and PD-L1 expression in tumor microenvironments. The validity of SIR indices as biomarkers in LARC requires further investigation.

## 1. Introduction

Rectal cancer constitutes approximately 35% of all colorectal cancers (CRCs). Its incidence in the European Union is estimated at 125,000 per year, and this is predicted to rise due to sociodemographic changes [1,2]. An alarming increase in the incidence of both colon and rectal cancers in young adults has been observed in recent years [3,4]. Prognoses, especially in advanced stages of the disease, remain unsatisfactory [5]. The current standard of care for patients with locally advanced rectal cancer (LARC) is neoadjuvant radiotherapy/chemoradiotherapy followed by surgery according to total mesorectal excision (TME) principles with or without postoperative chemotherapy [6,7,8]. However, the impact of such an approach on overall survival (OS) remains unclear, and it may cause long-term toxicities and impaired quality of life [9,10]. New markers are required to appropriately identify low- and high-risk patients, which is crucial for properly adjusting patients’ therapy. Blood-based systemic inflammatory response (SIR) markers such as LMR, NLR, and PLR are simple and cheap biomarkers with proven prognostic value in CRC [11,12,13,14]. However, the proper validation of these markers is lacking, and their roles in LARC are uncertain [15,16]. We conducted a prospective study on a well-selected group of patients with LARC. We investigated the reproducibility of the SIR markers, their correlations with clinicopathological characteristics, and their prognostic value.

## 2. Materials and Methods

A single-arm prospective study among patients treated at the Maria Skłodowska-Curie National Research Institute of Oncology in Warsaw was conducted. The eligibility criteria were as follows: (1) the patients were diagnosed with primary locally advanced rectal cancer confirmed by histopathology; (2) their clinical records, including demographic data and laboratory data, were available and complete; (3) the performance statuses of the patients were ECOG 0-2, and the patients had qualified to receive radio/chemoradiotherapy by multidisciplinary teams; and (4) the patients were >18 years old. The exclusion criteria were as follows: (1) the presence of distant metastasis at the time of diagnosis; (2) the presence of malignant tumors in other organs; (3) the presence of acute or chronic inflammatory diseases, hematological malignancies, autoimmune diseases, and other medical conditions that could affect inflammatory markers; and (4) prior immunosuppressive therapy. Blood samples from the patients were obtained three times within a median period of 21 days (range of 7–55 days). All the tests were performed prior to any oncological treatments. The differential white blood cell counts were analyzed using a Sysmex XN-550 hematology analyzer following the manufacturer’s protocol. The LMR, NLR, and PLR were calculated from the blood samples by dividing an absolute lymphocyte count by an absolute monocyte count, an absolute neutrophil count by an absolute lymphocyte count, and an absolute platelet count by an absolute lymphocyte count, respectively. The patients were divided in terms of the baseline values of their SIR markers into high and low LMR, NLR, and PLR groups. The cut-off values were determined based on our previous studies and the data available in the literature [17,18,19,20].

Formulas:LMR—absolute lymphocyte count (g/L)/absolute monocyte count (g/L)
NLR—absolute neutrophil count (g/L)/absolute lymphocyte count (g/L)
PLR—absolute platelet count (g/L)/absolute lymphocyte count (g/L)

All the patients received neoadjuvant radio/chemoradiotherapy according to the multidisciplinary teams’ decisions, which were based on the stage of the disease. Ten patients did not agree to proceed with surgery. Six patients progressed/proved to be inoperable before surgery. Surgery was performed on 44 patients. 

### 2.1. Histopathological Analysis

The post-surgical pathological results were collected and analyzed. There were 10 cases of complete pathological response (pCR). In two cases, no pathological specimens were available after surgery, and in three cases, the specimens were deemed not suitable for the histopathological analysis. Twenty-nine specimens were found suitable for the analysis. The presence of tumor-infiltrating immune cells in the tumor centers and the invasive margins was evaluated by immunohistochemistry using the antibodies for the CD8 antigen. For the immunohistochemical staining, primary monoclonal antibodies against CD8 (DAKO, Glostrup, Denmark, Cat. No IR623) with a DAKO EnVision FLEX detection system (DAKO, Denmark, Cat. No K8002) were used. Paraffin sections (4 μm on silanized slides) were deparaffinized, rehydrated, and then stained according to the manufacturer’s procedures. In a semi-quantitative assessment, a four-digit scale (0: 0–10% of the area of scarce and mild staining, 1: 11–50% of the area of moderate or intensive staining, 2: 50–75% of the area of intermediate or intensive staining, and 3: >75% of the area of intermediate or intensive staining) of the density of lymphocytes was used in the measurements for the tumor invasive margins. The inflammatory infiltrates containing lymphocytes, plasmacytes, monocytes/macrophages, and neutrophils were assessed histologically on H&E basic stain at the invasive fronts of the tumors using the same semi-quantitative four-digit scale. An example of intensive inflammatory infiltrates and scarce inflammatory infiltrates at the invasive margins is presented in Figure 1. Primary antibodies against MSH6 (DAKO, Denmark, Cat. No IR086) and PMS2 (DAKO, Denmark IR087) were used to detect the expression of microinstability indicator proteins. The percentage of positive cancer cells was estimated in each case, and the internal positive control consisted of lamina propria inflammatory cells and/or nontumoral glandular cells. As for the PD-L1 expression, clone 22C3 of the monoclonal antibody (DAKO, Denmark, Cat. No SK006) was used, and the staining was performed automatically in a closed system as supplied by the manufacturer. The expression was calculated as a CPS given the number of the PD-L1-staining cells (tumor cells, lymphocytes, macrophages) relative to all viable tumor cells, multiplied by 100% (the range of the results was between 0 and 100). An example of high and low expression of PD-L1-staining cells is presented in Figure 2. 

### 2.2. Statistical Analysis

The Shapiro–Wilk test was used to test the normality of the data distribution. The analysis of the repeatability of the measurements of SIR markers was evaluated using the Friedman test. Binomial variables were compared between measurements with the McNemar test. Additionally, confidence intervals for the proportions were calculated using a binomial exact calculation. Cohen’s Kappa was calculated to assess the extent of agreement between the first and the second measurements, including 95% confidence intervals. The relationships between parameters were assessed using Pearson’s correlation analysis. Statistical analyses were performed using the IBM SPSS Statistics ver. 23 software package and R software, version 4.0.5. The Kaplan–Meier procedure was performed to compare the survival and time without relapse between patients, with low and high levels of the LMR, NLR, and PLR. The log-rank test was used to verify whether any significant differences between groups were present. The 95% confidence intervals were calculated for a cumulative proportion of the patients who did not die/relapse. Correlations between qualitative or semi-qualitative variables were verified using Spearman’s correlation coefficients. The levels of the LMR, NLR, and PLR vs. the T, N, CR, and presence of progression were analyzed using Mann–Whitney U tests (comparison of 2 groups) or with a Kruskal–Wallis test (comparison of 3 groups), with a Dunn post hoc test.

### 2.3. Ethical Considerations

The study conformed to the provisions of the Declaration of Helsinki and was approved by the ethics committee of the National Institute of Oncology. All patients were informed of the investigational nature of this study and provided written informed consent.

## 3. Results

A total of 60 patients with rectal cancer treated at the Maria Skłodowska-Curie National Research Institute of Oncology in Warsaw between August 2017 and December 2020 were prospectively enrolled in the study. Forty-three males and seventeen females were included. The median age was 66.5 years (range of 29–89 years old). All the patients in the study were citizens of Poland of Caucasian ethnicity. The distributions of the cancer stages were as follows: stages II–IIIA, 8 (13%); stage IIIB, 41 (68%); and stage IIIC, 10 (17%). The stage of one of the patients remained undefined. There were no stage I or stage IV patients. All the rectal cancers were adenocarcinomas. The intermediate differentiation of the tumor was the most common—in 42 (70%) patients followed by the undefined differentiation—14 (23.3%). Two (3.3%) rectal cancers were well-differentiated (G1) and two (3.3%) poorly differentiated (G3). In terms of localization of the tumor within the rectum (distance of the lowest portion of the tumor from the anal verge), 28 (47%) patients had low, 24 (40%) middle, and 8 (13%) high rectal cancer. There were 15 (25%) smokers and 45 (75%) non-smokers. Most of the patients were overweight—23 (38%); 19 (32%) had normal weight; 17 (28%) were obese, and only 1 (2%) patient was underweight. Almost half of the patients (47%) had normal levels of carcinoembryionic antigen (<5.0 ng/mL). The characteristics of the patients are presented in Table 1.

The median values of the lymphocytes, monocytes, neutrophils, and platelet counts, as well as their ratios, are shown in Table A1.

### 3.1. Reproducibility

The patients were divided into high and low groups according to the baseline values of each SIR marker. The predetermined cut-offs were 2.6 for the LMR, 3.0 for the NLR, and 150 for the PLR. The numbers of patients who belonged to each group in each measurement are presented in Table A2. 

Over half of the patients (56.7%) (95% CI, 43.2–69.4%) were classified as LMR high, and 61.7% (95% CI, 48.2–73.9%) and 51.7% (95% CI, 38.4–64.8%) of the patients were assigned to the NLR low and PLR low groups accordingly. After the second measurements, 81.7% (95% CI, 69.6–90.5%) of the patients belonged to the same groups (LMR high or LMR low). In terms of the NLR and PLR, 73.3% (95% CI, 60.3–83.9%) and 78.3% (95% CI, 65.8–87.9%) of the patients were in the same groups, respectively. After three measurements, the percentages of patients who stayed in the same groups were nearly identical, as follows: 68.3% (95% CI, 55.0–79.7%) for the LMR and NLR and 70.0% (95% CI, 56.8–81.2%) for the PLR. For the LMR, NLR, and PLR, there were no significant changes in the percentages of the patients classified as low or high between all three measurements (*p* > 0.05 in all comparisons). The mean percentage change between the third and the first measurements of the lymphocytes, monocytes, neutrophils, and platelet counts ranged from −5.59% to 4.76%, and the standard errors ranged from 2.0 to 3.9 (Table 2).

The Cohen’s Kappa statistic for the extent of the agreement between the first and second measurements for the LMR was κ = 0.59 (95% CI, 0.39–0.79) (*p* < 0.001). For the NLR, the Kappa was κ = 0.45 (95% CI, 0.22–0.68) (*p* < 0.001), and for the PLR, κ = 0.53 (95% CI, 0.32–0.75) (*p* < 0.001), meaning in all cases, there was a moderate agreement between both measurements.

If the LMR at the first measurement was out of the range of 2.2–3.0 (±0.4 from the cut-off), then the risk of misclassification in the second measurement, defined as an affiliation to a different (high or low) group than initially, dropped to 5.0% (95% CI, 1.0–13.9%). In the case of the NLR, when it was outside of the range of 2.5–3.5 (±0.5) in the first test, it was 8.3% (95% CI, 2.8–18.4%), and in the case of a PLR outside of the range of 125–175 (±25), it was 10.0% (95% CI, 3.8–20.5%).

An analysis of the correlation between the first and third measurements of the LMR, NLR, and PLR was conducted. The LMR values were correlated with a coefficient of 0.776 (*p* < 0.00001). The NLR and PLR were correlated with coefficients of 0.696 (*p* < 0.000089) and 0.751 (*p* < 0.00001), respectively (Figure A1).

### 3.2. Correlation with Clinicopathological Characteristics

There was no significant correlation between the LMR, NLR, and PLR and the tumor size. There were no relationships between the pre-treatment levels of the SIR markers and both the progression and inoperability after neoadjuvant therapy as well as complete pathological responses. There were significant differences in the PLR levels between the N0, N1, and N2 subgroups (*p* = 0.033). A post hoc analysis confirmed that the PLR level in the N0 group was lower (116.35 (89.14–145.30) vs. N1, 147.27 (62.70–452.56); and vs. N2, 164.41 (93.47–321.83). There was no correlation between the LMR and the NLR, and the nodal involvement was observed (Table 3).

There was no significant correlation between the LMR, NLR, and PLR and the pre-treatment level of CEA (*p* > 0.05 in all cases) (Table A3). There was a significant positive correlation between the LMR and the cancer-related inflammatory infiltrates in the resected tissues (r = 0.38, *p* = 0.044) and the PD-L1 expression in the tumor cells and tumor-associated leukocytes (CPS) (r = 0.45, *p* = 0.016). The NLR and PLR were not related to the level of CPS or the inflammatory infiltrates. The correlation between the density of the CD8+ lymphocytes and the LMR, PLR, and NLR was not significant (Table 4).

The combined positive score was significantly positively correlated with the CD8+ (r = 0.56, *p* = 0.002), as well as with the inflammatory infiltrates (r = 0.51, *p* = 0.005) (Table A4). There was only one case of mismatch repair deficiency among the twenty-nine histopathologically assessed specimens (3.45%).

### 3.3. Prognostic Value 

The population of patients was analyzed in terms of recurrence-free survival (RFS) and OS depending on the pre-treatment levels of the LMR, NLR, and PLR.

### 3.4. Lymphocyte-to-Monocyte Ratio

The cumulative proportion of patients who did not relapse at the end of the observation period was 32% (95% CI = 8%; 100%) for the low LMR level group and 68% (95% CI = 53%; 87%) for the high LMR level group. The mean number of months without relapse was M = 39.03 for the low LMR level group and M = 47.01 for the high LMR level group (*p* = 0.641). At the end of the observation period, the cumulative proportion of alive patients was 80% (95% CI = 65%; 97%) for the low LMR level group and 80% (95% CI = 66%; 99%) for the high LMR level group. The mean time of survival was M = 44.81 months for the subjects with low LMR levels and M = 52.61 months for the subjects with high LMR levels (*p* = 0.597) (Figure 3).

### 3.5. Neutrophil-to-Lymphocyte Ratio

The mean number of months without relapse for the patients with low NLR levels was M = 48.79, and for the patients with high NLR levels, it was M = 36.91. The cumulative proportion of subjects who did not relapse at the end of the observation period was 71% (95% CI = 57%; 90%) for the low NLR level group and 30% (95% CI = 7%; 100%) for the high NLR level group (*p* = 0.225). No differences were detected between the survival times of the patients with low and high NLR levels (*p* = 0.927). The mean time of survival was M = 51.36 months for the subjects with low NLR levels and M = 45.66 months for the subjects with high NLR levels. The cumulative proportion of alive patients at the end of the follow-up period was 76% (95% CI = 59%; 98%) for the low NLR level group and 83% (95% CI = 69%; 100%) for the high NLR level group (Figure 4).

### 3.6. Platelet-to-Lymphocyte Ratio

The cumulative proportion of patients who did not relapse at the end of the observation period was 63% (95% CI = 46%; 86%) for the low PLR level group and 47% (95% CI = 20%; 100%) for the high PLR level group. The mean number of months without a relapse was M = 40.86 among the patients with low PLR levels and M = 44.48 among the patients with high PLR levels (*p* = 0.869). The mean time of survival was M = 42.57 months for the patients with low PLR levels and M = 54.56 for the patients with high PLR levels. The cumulative proportion of alive subjects was 72% (95% CI = 56%; 94%) for the low PLR level group and 89% (95% CI = 78%; 100%) for the high PLR level group (*p* = 0.261) (Figure 5).

## 4. Discussion

Cancer may induce both local and systemic inflammatory reactions [21]. The LMR, NLR, and PLR are blood-based biomarkers of cancer-related inflammation. In our study, we proved that there was a strong correlation between the LMR and cancer-related inflammatory infiltrates in the resected tissues. Similar results have been reported for cholangiocarcinoma, colorectal, and breast cancers [22,23,24]. However, no correlation between the SIR markers and tumor-infiltrating CD8 lymphocytes was found, which was in line with other studies on both rectal and left-sided colon cancers [25,26]. This apparent discrepancy may have been due to the large populations of neutrophils, macrophages, or other subsets of lymphocytes in the inflammatory infiltrates. We found a correlation between the LMR and PD-L1 expression in the tumor cells and tumor-associated leukocytes relative to all the viable tumor cells (CPS). To our knowledge, these are the first data on a correlation between the SIR markers and CPS in colorectal cancer. In other malignancies, the data on correlations between SIR markers and PD-L1 expression are conflicting [27,28]. We found no association between the SIR markers and the level of CEA, which corresponded to a retrospective study on rectal cancer patients [29]. The PD-L1 expression in the immune cells was positively correlated with both the inflammatory infiltrates and the tumor-infiltrating CD8 lymphocytes. Similar relationships have been reported in hepatocellular carcinoma, cholangiocarcinoma, and colorectal cancer [30,31,32]. The LMR, NLR, and PLR are biomarkers with high prognostic value in many malignancies. However, their roles in LARC are not clear and lack proper validation. The number of studies assessing their reproducibility is very limited. To the best of our knowledge, our study is the first to directly investigate this subject in a prospectively enrolled cohort. Reference and cut-off values for the SIR markers are not well-established. According to analyses of ostensibly healthy populations, the average values of the LMR, NLR, and PLR may differ depending on race, sex, and age. The mean values for the LMR in healthy individuals were significantly higher, and the mean values for the NLR and PLR were lower in comparison to our results [33,34,35]. Our findings were based on a well-selected group of patients with untreated LARC with no concomitant acute or chronic diseases that could have influenced the levels of inflammatory markers, which suggests that all three SIR markers are only moderately reproducible. When divided into high and low groups, the percentages of patients who stayed in the same groups after three measurements were nearly the same for all the parameters (68.3% for the LMR and NLR and 70% for the PLR). Nearly one-third of the patients’ affiliations with a group changed between the assessments. However, if the first measurement was out of the range of approximately ±15% from the cut-off, the risk of misclassification in the second measurement dropped significantly, and in terms of the LMR, this dropped to 5% (95% CI, 1.0–13.9%), while for the NLR, it dropped to 8.3% (95% CI, 2.8–18.4%), and for the PLR, it dropped to 10% (95% CI, 3.8–20.5%). These results were in line with our previous retrospective study on the reproducibility of the LMR in patients with LARC, where two peripheral blood tests within five weeks prior to beginning anti-cancer therapies were performed [20]. The stability of the NLR over time, up to 100 days, has been demonstrated in cardiac surgery patients; however, it has not been confirmed in a cancer population [36]. No other studies investigating the reproducibility of SIR markers have been found in the literature. We analyzed the RFS and OS of patients depending on the levels of their LMR, NLR, and PLR. We found no statistically significant correlations in terms of RFS and OS between the high and low LMR, NLR, and PLR groups. These results were not consistent with the majority of studies assessing the whole population of CRC [37,38,39,40]. However, among trials restricted to LARC, the impacts of SIR markers on recurrences and survival have been conflicting. Wu et al. showed no correlation between the LMR and the DFS or OS in a non-metastatic rectal cancer population [15]. Similarly, in a large study of over 1500 LARC patients by Dudani et al., no statistically significant correlation between the NLR, PLR, and DFS and the OS was proven [16]. These findings were supported by the results of the study by Ishikawa and Portale et al. [41,42]. Most meta-analyses have suggested that the SIR markers in CRC have prognostic value, and these have included patients with both metastatic and non-metastatic disease [43,44]. The association between SIR markers and prognosis was less noted in non-metastatic stages. There are data that have indicated that SIR markers are associated with adverse OS in colon cancer but not in rectal cancer [45]. Our results confirmed that the prognostic value of the SIR markers in LARC is less evident than those among the whole CRC population. The phenomenon of cancer-related inflammation is important for understanding the roles of SIR markers. The relationship between cancer and inflammation has been investigated since the 19th century when Virchow first observed that cancer tends to originate from chronically inflamed sites [46]. Through the recruitment of inflammatory cells and cytokines, the production of reactive oxygen species, and the inhibition of repair programs, inflammation promotes the uncontrolled proliferation of defective cells and potentiates neoplastic risk. Inflammatory cells are abundant in a tumor’s microenvironment [47]. They reflect a reaction of the host towards a tumor, but they also serve as a product of cancer-related cells and a tumor’s predisposition toward invading and suppressing the immune system [48]. Lymphocyte counts reflect systemic inflammatory responses by inducing the production of anti-tumor cytokines, and cytotoxic activity suppresses a cancer’s proliferation and spread [49]. Monocytes, on the contrary, have proven to contribute to a tumor’s progression and metastatic activity [50]. Neutrophils, accounting for 50–70% of leukocytes, play a central role in cancer-related inflammation. Releasing reactive oxygen and nitrogen species that damage DNA, they play a substantial role in cancer initiation [51]. Tumor progression is boosted by neutrophil-derived chemokines and cytokines that mediate the process of angiogenesis [52]. Neutrocytes take part in suppressing T-lymphocyte proliferation, reducing the anti-tumoral effect of NK cells and promoting metastatic spread [53,54]. Similarly, platelets, by releasing cytokines and growth factors, contribute to carcinogenesis. There is a substantial interaction between thrombocyte activation and cancer progression. Tumor cells produce cytokines, such as IL-6, that stimulate thrombocytosis. In turn, thrombocytes promote further tumor growth, leading to an even more intensive stimulation and activation of platelets [55]. These immunological interactions have led to the introduction of SIR markers and the investigation of their potential roles in clinical practice. Our study revealed interesting aspects of the SIR markers in LARC. 

There were two main limitations concerning our study: (a) it had a relatively small group of patients, and (b) our studied population was homogenous, consisting entirely of Caucasian citizens of Poland. Moreover, other factors might have had an impact on the results of our study such as the lack of well-defined cut-off values for the LMR, NLR, and PLR and the possible influence of other parameters (e.g., age, sex, comorbidities, smoking) on the level of the LMR, NLR, and PLR. The time between measurements of blood samples varied, which might have had an impact on the blood results. Finally, the immunohistochemical data may suffer a bias due to the fact that several patients either did not proceed with surgery or had complete pathological responses. Future studies should include larger and mixed populations to confirm our results. Despite its limitations, our study explored subjects that are rarely present in the literature. A better understanding of the roles of SIR indices in LARC and their relationships with other clinicopathological features may enable the application of these markers in clinical practice.

## 5. Conclusions

The LMR, NLR, and PLR are peripheral blood-based markers of cancer-related inflammation. Our results suggest that the LMR is correlated with inflammatory infiltrates and PD-L1 expression in a tumor’s microenvironment. However, the prognostic value of the SIR markers appears to be less evident among the patients with LARC compared to other colon cancers and most other malignancies, with no statistically significant impact on the RFS or OS in our study. The reproducibility of the SIR markers is moderate. More prospective studies are required to assess the validity of the SIR indices as biomarkers in LARC.

## Figures and Tables

**Figure 1 biomedicines-11-03048-f001:**
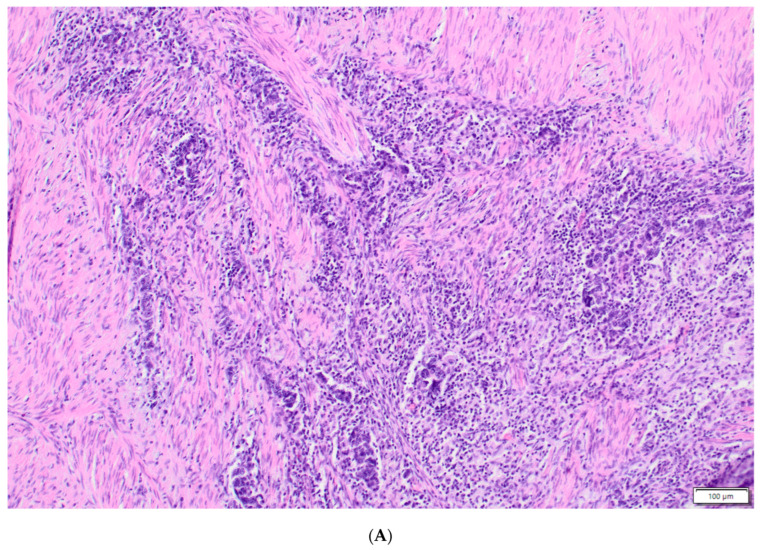
Inflammatory infiltrates at the invasive margins of the cancers. The intensive inflammatory infiltrate (**A**) versus nearly no inflammatory cells (**B**) at the invasive margins of the tumors (both H&E ×100).

**Figure 2 biomedicines-11-03048-f002:**
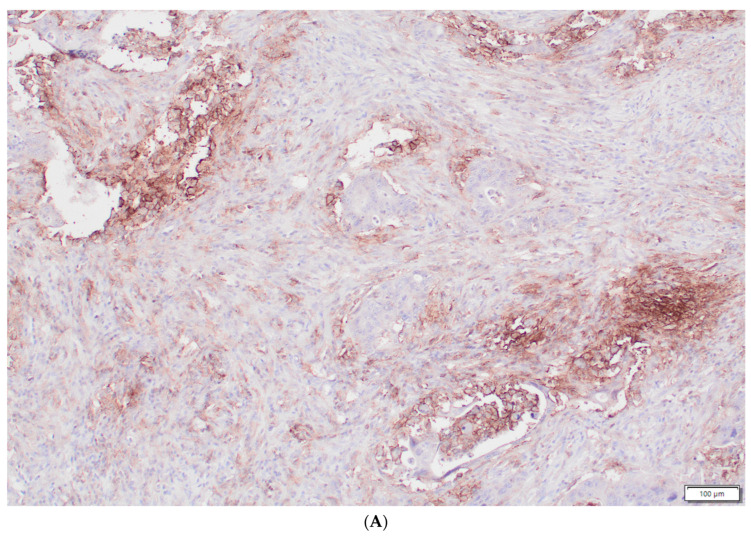
PD-L1-staining cells at the invasive margins of the cancers. The high expression of PD-L1-staining cells (**A**) versus nearly no PD-L1-staining cells (**B**) at the invasive margins of the tumors (DAKO 22C3 antibody).

**Figure 3 biomedicines-11-03048-f003:**
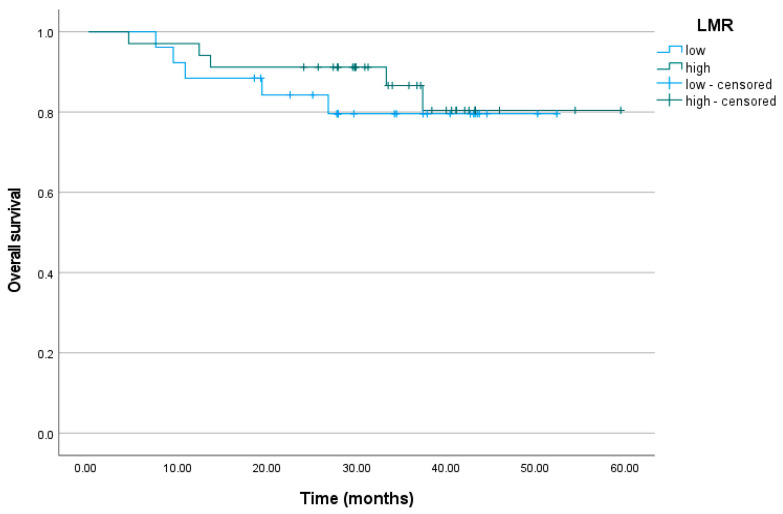
Overall survival curve for the patients with low and high LMR levels. LMR, lymphocyte-to-monocyte ratio.

**Figure 4 biomedicines-11-03048-f004:**
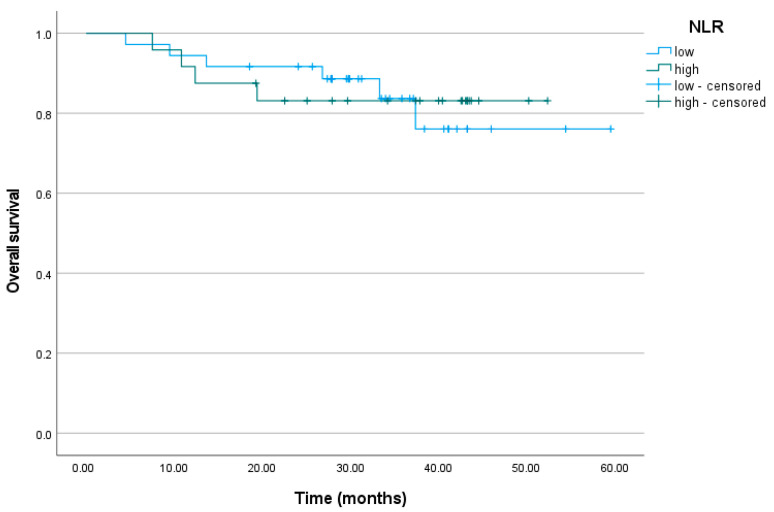
Overall survival curve for the patients with low and high NLR levels. NLR, neutrophil-to-lymphocyte ratio.

**Figure 5 biomedicines-11-03048-f005:**
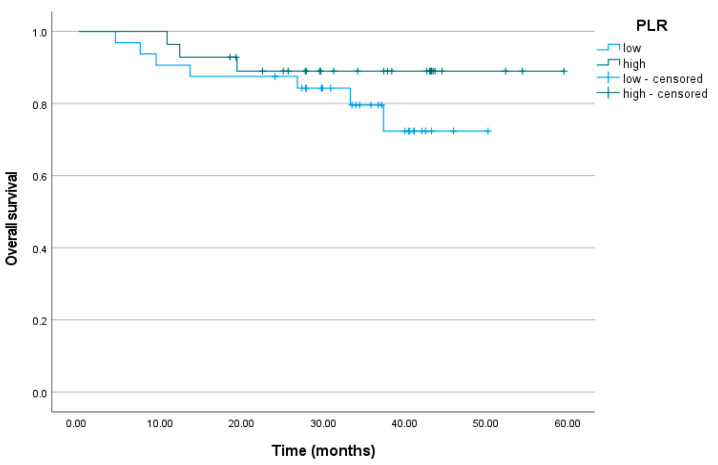
Overall survival curve for the patients with low and high PLR levels. PLR, platelet-to-lymphocyte ratio.

**Table 1 biomedicines-11-03048-t001:** Characteristics of the patients.

	All Patients (*n* = 60)
Age (years), median (range)	66.5 (29–89)
Sex, *n* (%)	
Male	43 (71.7)
Female	17 (28.3)
BMI, *n* (%)	-
<18.5	1 (2)
18.5–25	19 (32)
25–30	23 (38)
≥30	17 (28)
Smokers, *n* (%)	15 (25)
Non-smokers, *n* (%)	45 (75)
CEA (ng/mL), median (range)	21.89 (0.86–69.96)
Normal level (<5.0 ng/mL), n (%)	28 (47)
Elevated level (≥5.0 ng/mL), n (%)	32 (53)
Tumor, *n* (%)	
T3	55 (91.7)
T4	5 (8.3)
Lymph nodes, *n* (%)	
N0	8 (13.3)
N1	35 (58.3)
N2	16 (26.7)
Nx	1 (1.7)
Grade, *n* (%)	
G1	2 (3.3)
G2	42 (70)
G3	2 (3.3)
Gx	14 (23.3)
Stage, *n* (%)	
II–IIIA	8 (13.3)
IIIB	41 (68.3)
IIIC	10 (16.7)
Tumor localization, *n* (%)	
Low rectum	28 (47)
Middle rectum	24 (40)
High rectum	8 (13)
Time between measurements (days), median (range)	
1st–2nd	9 (1–42)
2nd–3rd	11 (1–34)
1st–3rd	21 (7–55)

BMI, body mass index; CEA, carcinoembryonic antigen.

**Table 2 biomedicines-11-03048-t002:** Calculations of the percentages of the changes between the third measurements vs. the first measurements.

% Change	*n*	Mean	Standard Deviation	Standard Error	Median	Minimum	Maximum
L	60	4.76	30.23	3.9	0.75	−60.61	92.86
M	60	3.88	24.39	3.1	4.78	−40.00	85.71
N	60	−5.59	20.57	2.7	−8.20	−47.70	43.66
WBC	60	−2.39	17.28	2.2	−3.86	−39.78	42.12
PLT	60	1.29	15.30	2.0	−0.70	−29.32	44.60

L, lymphocytes; M, monocytes; N, neutrophils; WBC, white blood cells; PLT, platelets.

**Table 3 biomedicines-11-03048-t003:** Average value of the LMR, NLR, and PLR depending on the size of the tumor, nodal status, complete pathological response, and presence of progression after neoadjuvant treatment.

	LMR Avg *		NLR Avg *		PLR Avg *	
	Median (Range)	*p*-Value	Median (Range)	*p*-Value	Median (Range)	*p*-Value
T						
T3	2.94 (1.12–6.91)	0.470	2.71 (1.01–6.90)	0.336	142.46 (62.70–452.56)	0.377
T4	2.49 (1.19–3.84)		3.10 (2.07–8.92)		185.88 (97.58–390.89)	
N						
0	3.52 (1.50–5.34)	0.714	2.27 (2.06–4.07)	0.457	116.35 (89.14–145.30)	0.033 (0 vs. 1 and 2)
1	2.91 (1.14–6.91)		2.91 (1.01–8.92)		147.27 (62.70–452.56)	
2	147.27 (62.70–452.56)		2.68 (1.94–5.44)		164.41 (93.47–321.83)	
CR						
No pCR	3.02 (1.12–6.91)	0.867	2.71 (1.01–6.90)	0.796	136.65 (62.70–392.60)	0.309
pCR	2.85 (1.80–5.23)		2.56 (1.91–6.39)		153.55 (93.47–452.56)	
Progression/inoperability						
No	2.86 (1.12–6.91)	0.990	2.71 (1.01–8.92)	0.805	144.43 (62.70–452.56)	0.931
Yes	2.98 (1.14–3.86)		2.72 (1.76–6.15)		149.28 (95.91–349.34)	

LMR, lymphocyte-to-monocyte ratio; NLR, neutrophil-to-lymphocyte ratio; PLR, platelet-to-lymphocyte ratio; pCR, pathological complete response; *, average level from all three measurements with analyses using the Mann–Whitney U test (T; progression/inoperability, CR) or the Kruskal–Wallis test (N).

**Table 4 biomedicines-11-03048-t004:** Correlation between the LMR, NLR, and PLR and the CPS, CD8+ lymphocytes, and inflammatory infiltrates.

	CPS		CD8+		Inflammatory Infiltrate	
	R	*p* Value	R	*p* Value	r	*p* Value
LMR avg *	0.45	**0.016**	0.21	0.266	0.38	0.044
NLR avg *	−0.19	0.316	−0.08	0.691	−0.15	0.447
PLR avg *	0.16	0.401	0.06	0.744	0.09	0.626

LMR, lymphocyte-to-monocyte ratio; NLR, neutrophil-to-lymphocyte ratio; PLR, platelet-to-lymphocyte ratio; CPS, combined positive score; *, average level from all three measurements; r, Spearman’s correlation coefficient.

## Data Availability

The data presented in this study are available on request from the corresponding author. The data are not publicly available due to medical data privacy issues.

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
