# Peer review of "A Prospective Study on the Roles of the Lymphocyte-to-Monocyte Ratio (LMR), Neutrophil-to-Lymphocyte Ratio (NLR), and Platelet-to-Lymphocyte Ratio (PLR) in Patients with Locally Advanced Rectal Cancer"

_biomedicines, 2023, doi:10.3390/biomedicines11113048_

Round 1
Reviewer 1 Report
Comments and Suggestions for Authors
Comments:
1. Please show the representative images of NLR, PLR, and LMR.
2. Please show representative image of PD-L1 IHC.
3. Add scale bar on Figure 1.
4. On Table 1, add patients' BMI, smoking and alcohol history, CEA level.
Reviewer 2 Report
Comments and Suggestions for Authors
1- Provide number and demographic information of patients in the text.
2- More literature needed in the introduction section.
3- Please clarify the necessity of the work.
4- I confused about the contents of tables and their information. Please presented them as a scientific work for readers.
5- Moderate editing of English language grammar and spelling is required.
Comments on the Quality of English LanguageModerate editing of English language grammar and spelling is required.
Reviewer 3 Report
Comments and Suggestions for Authors
The article presents an original research regarding the role of systemic inflammatory biomarkres ( NLR, LMR and PLR) in predicting outcomes after rectal cancer surgery. The article is of novelty and the statistic methods used are appropriate.
Some minor issues could be improved:
1. More data regarding the type of rectal cancer would be interesting for the reader.
2. A paragraph with limitation of the study should be added.